# Free-breathing phase-resolved functional lung (PREFUL) low-field magnetic resonance imaging (LF-MRI) of pulmonary dysfunction after surviving childhood cancer

## Abstract

**Background** Childhood cancer survivors have a high risk of chronic multi-organ disease that does not plateau over time. To date, there is a lack of sensitive diagnostic techniques that allow early detection of tissue damage before clinical symptoms occur, particularly regarding pulmonary function. Free-breathing phase-resolved functional lung (PREFUL) low-field magnetic resonance imaging (LF-MRI) may enable visualization and quantification of functional and structural lung damage without the need of specific contrast agents.
**Methods** In this single-center, cross-sectional diagnostic study, we performed LF-MRI in a cohort of $n = 27$ children and adolescents (age range: 5 to 17 years) after treatment for acute lymphoblastic leukemia (ALL; $n = 21$) and Hodgkin's disease (HD; $n = 6$) to determine the frequency of morphologic and functional lung parenchymal changes.
**Results** Here, we show that despite the absence of clinical symptoms, significant time-dependent pulmonary ventilation and perfusion defects are detected. A negative correlation between the time after the end of therapy and defect-free lung tissue in the cohort of patients treated for ALL (Spearman-coefficient = − 0.69, $p = 0.0005$) is observed.
**Conclusions** Our results suggest an increase in pulmonary ventilation and perfusion defects preceding the increase in chronic disease that has already been reported in this patient population. Further research is needed to determine whether the functional abnormalities described in this study are an early morphological correlate of developing organ damage that may become clinically evident over time. PREFUL MRI may be an effective and highly sensitive tool for early detection of these changes in lung function, allowing longitudinal studies for risk stratification and potential future treatment adaptation.

## Plain language summary

Treatment of children and adolescents with cancer has led to a steady improvement in cure rates. At the same time, our knowledge of the possible long-term effects of conventional chemotherapy-based treatment in children remains limited. In this study, we performed a type of scan called PREFUL MRI in children and adolescents after treatment for acute lymphoblastic leukemia and Hodgkin's disease that did not require treatment with chemicals called contrast agents or ionizing radiation. Our results show that following treatment there is an increasing and time-dependent dysfunction in the lungs, despite there being no clinical symptoms. Our imaging method might be an effective and highly sensitive tool for early detection of long-term toxicity after cancer treatment. It could also be used to monitor the side effects of new cancer treatments.

Currently, overall cure rates of therapy for childhood acute lymphoblastic leukemia (ALL) and Hodgkin's disease (HD) exceed 80%[1]. Apart from the development of supportive measures and novel targeted therapies, this success is still based largely on the optimized and risk-adapted dosing and scheduling of conventional chemotherapeutic agents and the addition of radiotherapy in cases with HD and suboptimal response to induction therapy. Even though treatment regimens are increasingly risk-adapted and therefore less intensive than previous therapies, they are still associated with organ toxicities such as cardiac dysfunction, osteonecrosis, neurocognitive impairment, and secondary malignant neoplasms[2]. Cardiovascular diseases are common late effects. Likewise, pulmonary diseases are reported as chronic health conditions among childhood cancer survivors and are associated with increased lifelong morbidity and mortality[3–5].

✉e-mail: alexander.dierl@uk-erlangen.de

Known risk factors for emerging pulmonary dysfunction are oxidative stress due to lung-toxic chemotherapeutics and free radical formation during radiotherapy or pulmonary surgery[6,7]. Spirometry and body plethysmography are widely available and are considered the standard of care for examining lung function. However, these techniques bear limited sensitivity. Lung scintigraphy measures both pulmonary ventilation and perfusion[8,9]. This method requires the administration of radioactive tracers and must therefore be used restrictively, particularly in vulnerable patient populations such as children and young adults. Advances in morphological and functional MR imaging, when appropriately indicated, have recently expanded the range of applications to include examinations of lung tissue without inferiority to CT scans[10]. At low field strength, this approach has enhanced the imaging quality near air-tissue interfaces without the need for ionizing radiation[11,12].

To date, there is a lack of studies on the non-invasive assessment of possible long-term pulmonary damage after chemotherapy in children and adolescents, as well as in adults. Therefore, we performed a cross-sectional investigator-initiated trial to evaluate morphological lung changes in children and adolescents after oncological treatment for ALL or HD at a single academic medical centre using free-breathing phase-resolved functional lung (PREFUL) MRI on a low-field system to identify indicators for potentially emerging pulmonary toxicity after treatment for childhood ALL and HD. Our results suggest an increase in pulmonary dysfunction similar to the increase in chronic disease that has already been reported in this patient population. PREFUL-MRI in a low-field system might be an effective tool for early detection of long-term toxicity after oncological treatment in childhood and adolescence.

## Methods
### Study design
We performed a cross-sectional, investigator-initiated observational trial at a single center to evaluate morphological lung changes in children and adolescents after oncological treatment of ALL or HD at a single academic medical centre. The study was registered with ClinicalTrials.gov, US National Library of Medicine (Identifier No: NCT06093334) on 16/10/2023. The ethics committee of the University of Erlangen-Nuernberg granted approval for the study (Number of ethics vote: 23-472-B). All parents or guardians gave written informed consent for their children to participate in the study.

### Objectives
The primary objective was the determination of the frequency of morphologic changes of lung parenchyma at low-field-strength proton MRI. Secondary outcomes included functional lung changes comprising ventilation defects (whole-lung ventilation defect percentage [VDP]), perfusion defects (whole-lung perfusion defect percentage [QDP]), the match (ventilation-perfusion [V/Q] match), defect of both (whole-lung V/Q defect [V/Q] defect), and reported clinical symptoms.

### Patients and sample size
Between October 2023 and December 2024, 27 patients aged 5–17 years were recruited for the study. The baseline characteristics and MRI results of the patients are given in Table 1. A total of 21 patients after completion of intensive therapy for ALL were included. In addition, six patients after therapy for HD were evaluated. In both study groups, the inclusion criteria required a patient age between 5 and 17 years. Exclusion criteria were the impossibility of performing the MRI (e.g., electrical implants such as cardiac pacemakers or perfusion pumps, etc.) or performing the MRI without sedation, the impossibility of physical exertion (e.g., after serious operations or post-operative deformities) and poor or unstable general condition or pregnancy.

### Study flow
Participants were assessed for current medication and medical history, specifically for the presence of respiratory complaints or frequent pulmonary infections. A physical examination, specifically including auscultation of the lungs, was performed. After a rest period, the participants underwent low-field-strength MRI.

### MRI protocol
All participants underwent morphological and functional free-breathing lung MRI at low field strength. This MRI method records signal changes in the lungs over several respiratory and cardiac cycles while the patient breathes freely. A high temporal resolution is achieved without the need for breath stops. By resolving the cycles, the pulmonary ventilation and perfusion in metric units can then be calculated from the imaging data (Fig. 1). The parameters for the different sequences are presented in Table 2[13–18]. Morphological MRI was evaluated by a radiologist, and functional sequences were analyzed using dedicated research software (MR Lung version 2.2.0, Siemens Healthineers, Forchheim, Germany)[17].

The following data were calculated: normalized perfusion was expressed as a percentage relative to a full-blood signal region[15]. Regional ventilation expressed as a percentage of signal strength, was computed using the formula $\frac{S_{mid}}{S_{insp}} - \frac{S_{mid}}{S_{exp}}$, where $S$ represents the signal intensity at end-inspiration ($S_{insp}$), end-expiration ($S_{exp}$), and the mid-respiratory position ($S_{mid}$)[19]. Flow-volume loop correlation was determined by correlating the flow-volume loop (derived from the reconstructed ventilation cycle) with an automatically determined reference healthy region, defined as the largest connected region within the 80th to 90th regional ventilation percentiles[20]. From these parameter maps, the percentage of defect areas for perfusion (QDP) and ventilation (VDP) was determined.

The percentage of regions exhibiting concurrent defects in both perfusion and ventilation (V/Q defect), exclusive defects in either perfusion or ventilation, and non-defects regions (V/Q match) was quantified based on normalized perfusion and flow-volume loop correlation metrics.

### Statistics and reproducibility
All statistical analyses were performed using GraphPad Prism (version 7.00 or later, GraphPad Software, La Jolla, CA, USA). Continuous variables are given as means with SDs and categorical variables as numbers with percentages. The results of the rank correlation measurement are reported as Spearman's rho (rs). An independent sample $t$-test (two-tailed) is used to determine significant differences between two groups. In cases of violated normal distribution the Mann–Whitney $U$-Test (two-tailed) is applied. In all statistical analyses, $p < 0.05$ is considered to indicate statistically significant difference in all analyses. The sample size used for analysis was $n = 27$. To validate our imaging approach, three patients with ALL were included in the study before consecutive allogeneic stem cell transplantation and examined again after successful transplantation. These repeated examinations were used for internal validation of the MRI measurements, as they were performed in a close temporal interval (112 up to 386 days) on the same participants.

## Results
### Respiratory problems are not commonly reported by pediatric cancer survivors
Of the 27 patients who underwent low-field MRI, only one participant in the ALL group and one in the HD group showed any morphological changes of the lung tissue. One patient in the ALL group exhibited linear atelectasis, while one patient in the HD group had dystelectasis of the apical right lower lobe (Fig. 2). None of these patients showed any clinical symptoms. Neither the patients themselves nor their parents reported reduced physical fitness. One patient in the ALL group complained of a chronic cough without other restrictions on the day of the examination. None of the study participants reported smoking.

### PREFUL MRI reveals subclinical pulmonary ventilation and perfusion defects
Participants who completed the intensive treatment for ALL but still received chemotherapy according to the ALL BFM 2017 (EudraCT: AIEOP-

**Table 1 | Clinical characteristics and functional MRI results of the study participants**

| Characteristic | ALL-Group (n = 21) | HD-Group (n = 6) |
|---|---|---|
| Age (y) | 11.1 ± 2,5 [5−17] | 14 ± 2.9 [9−17] |
| Height (m) | 1.5 ± 0.2 [1.1−1.81] | 1.6 ± 0.2 [1.371.85] |
| Weight (kg) | 37.6 ± 12.7 [20−66] | 53.2 ± 19.8 [22.4−78] |
| Sex[c] | | |
| Male | 14 (66.7%) | 6 (100%) |
| Female | 7 (33.3%) | 0 |
| Oncological treatment[c] | | |
| Only Chemotherapy | 18 (85.7%) | 3 (50%) |
| Radiochemotherapy[a] | 0 (0%) | 3 (50%) |
| Chemotherapy and consecutive allogeneic SCT (including TBI) | 3 (14.3%) | 0 |
| Study group[c] | | |
| Completed intensive therapy[b]/in therapy | 9 (42.9%) | 0 |
| Completed oncological treatment | 12 (57.1%) | 6 (100%) |
| Mean time point after beginning of therapy (days) | | |
| Completed intensive therapy[b]/in therapy | 739.3 ± 485.9 [203−1972] | 759.3 ± 439.1 [346−1657] |
| Completed oncological treatment | 2621.5 ± 978.5 [881−4243] | |
| Mean time point after beginning of therapy (years) | | |
| Completed intensive therapy[b]/in therapy | 2.03 ± 1.33 [0.56−5.4] | |
| Completed oncological treatment | 7.18 ± 2.68 [2.41−11.62] | 2.08 ± 1.2 [0.95−4.54] |
| Symptoms reported[c] | | |
| None | 20 (95.2%) | 6 (100%) |
| Chronic cough | 1 (4.8%) | 0 (0%) |
| VQ-Non defect (%) | | |
| Completed intensive therapy[b]/in therapy | 96.3 ± 3.0 [90.8−100] | |
| Completed oncological treatment | 85.8 ± 10.7 [76.1−99.4] | 85.1 ± 7.1 [74.2−92.9] |
| VDP (%) | | |
| Completed intensive therapy[b]/in therapy | 3.4 ± 2.9 [0−8.4] | 14.4 ± 6.6 [7−23.7] |
| Completed oncological treatment | 11.4 ± 7.3 [0.6−20.6] | |
| QDP (%) | | |
| Completed intensive therapy[b]/in therapy | 0.4 ± 0.7 [0 − 2.2] | 0.5 ± 0.8 [0−2.1] |
| Completed oncological treatment | 2.8 ± 5.4 [0−18.9] | |

Unless otherwise specified, data are means ± SDs. The values in the square brackets provide the range from the minimum to the maximum value.

*QDP* perfusion defect percentage, *VDP* ventilation defect percentage, *V/Q* ventilation-perfusion, *TBI* total body irradiation, *SCT* stem cell transplantation.

[a]Lung tissue contained in the radiation field.

[b]Still undergoing maintenance therapy according to ALL BFM 2017 therapy protocol.

[c]Data are numbers of participants, with percentages in parentheses.

BFM 2017 2016-001935-12) therapy protocol (i.e., maintenance therapy including oral Methotrexate and Mercaptopurine) were assigned to the "in therapy" group (n = 9). The remaining participants (n = 18) had already completed the entire oncological treatment for ALL or HD with a time interval of 2 to a maximum of 11 years since the start of therapy ("therapy completed"). Figure 3 demonstrates representative functional MRI images

of lung ventilation and perfusion of 5 study participants at different time points after the start of oncological treatment.

Figure 4a demonstrates the comparison of the functional MRI parameters of ALL patients at different timepoints in the oncological treatment. Children in the "in therapy" group had statistically significant higher V/Q-match (96,29%, p = 0.0137) and less VDP (3.40%, p = 0.0079) compared to patients in the "after therapy" group (V/Q-match = 85.82%, VDP = 11.40%). In addition, these patients showed larger, but not statistically significant QDP (p = 0.14).

To investigate associations between the functional MRI parameters and the time span after initial start of the therapy in the cohort of ALL patients, we performed Spearmans's correlation analysis. Here we could demonstrate statistically significant correlation between the time span after start of the treatment and increased VDP (rs = 0.6628, p = 0.0011) as well as increased QDP (rs = 0.6152, p = 0.0030). An inverse and statically significant correlation between V/Q-match and the time span was revealed (rs = −0.6901, p = 0.0005). The results of the correlation analysis are presented in Fig. 4b.

As observed in the ALL cohort, study participants in the group of patients with HD exhibited reduced V/Q match values increasing with time after therapy initiation. However, these results did not reach statistical significance. These effects were particularly pronounced in the two patients with the longest interval between therapy initiation and the examination date. When comparing Hodgkin's lymphoma patients with and without radiation treatment, no statistically significant differences regarding V/Q-match (p = 0.9438), VDP (p = 0.8782) and QDP (p > 0.99) were found. The results are presented in the Supplementary Data file 2.

Three patients after treatment for ALL performed the MRI examination twice. Here, the measurements before and after allogeneic stem cell transplantation showed only minor deviations in the functional MRI parameters. This indicates valid data collection without the influence of major uncontrollable confounding variables (as we do not expect any additional late damage due to side effects shortly after allogeneic stem cell transplantation). The results of the repeated examinations are presented in Table 3.

## Discussion

Recently, there have been some encouraging pilot projects and initiatives to improve long term follow up care in Germany (e.g., Projects VIVE, LE-Na, AELKI), but there is a lack of studies using novel highly sensitive imaging techniques to objectively record and monitor organ damage after childhood cancer. Long term follow-up care is already recommended for survivors[21]. In contrast, pulmonary functional testing is not routinely performed except after hematopoietic stem-cell transplantation (HSCT). Leading experts underline the importance of screening for pulmonary diseases in long term follow up care. They also highlight the current paucity of evidence, reveal relevant knowledge gaps and emphasize that collaborative studies and reports are urgently needed[22]. The use of PREFUL MRI has proven to be capable of detecting ventilation and perfusion defects in patients after Coronavirus disease (COVID 19)[13]. In our study population, morphological MRI findings in form of linear atelectasis and dystelectasis were described in only two cases. The particular advantage of using PREFUL MRI is the possibility of a functional assessment of pulmonary ventilation and perfusion dynamics. In contrast to the inconspicuous primary image data, we detected severe time-dependent changes in functional lung assessment. In our cohort, patients after completed treatment for ALL showed significantly greater ventilatory defects than patients who were still undergoing oncologic treatment (Figs. 3 and 4). Furthermore, we identified a significant correlation between the time since the start of therapy and the occurrence of ventilation deficits (Fig. 4). MRI data of individuals treated for HD also showed time-dependent pulmonary defects (Table 1). This cohort of Hodgkin's disease patients consisted of only 6 individuals, 3 of whom received radiation therapy. Nevertheless, we found a decrease of defect free lung tissue similar to the defects of ALL-patients, who already completed therapy. Our results therefore suggest that the described changes in

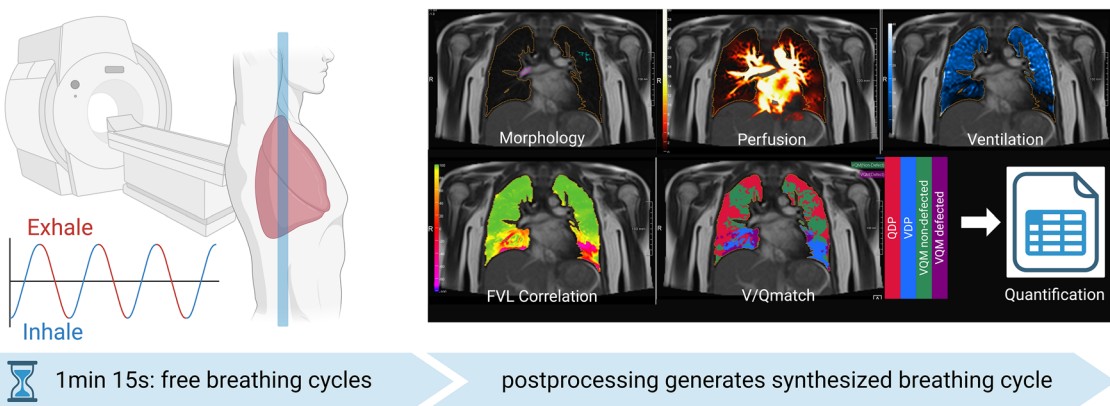

**Fig. 1 | Principle of free-breathing phase-resolved functional lung (PREFUL) MRI on a low-field system.** Signal changes of respiratory and cardiac cycles in free breathing patients are recorded and functional pulmonary parameters (Perfusion Map and Ventilation Map) are generated. Created with BioRender.com.

## Table 2 | PREFUL-MRI parameters and sequences

| MRI and sequence parameters | bSSFP | PROPELLER | |
|---|---|---|---|
| Orientation | Coronal | Transversal | Coronal |
| Fat-saturation | None | None | STIR[a] |
| Repetition time (ms) | 231.3 | 2000 | 2630 |
| Echo time (ms) | 1.3 | 35 | 80 |
| in-plane resolution (mm²) | 2 × 2 | 1.3 × 1.3 | 1.4 × 1.4 |
| Slice thickness (mm) | 15 | 6 | 5 |
| Matrix | 256 × 256[b] | 304 × 272 | 272 × 272 |
| Parallel imaging acceleration factor | 2 | 2 | 2 |
| Number of time points | 388 | Not applicable | |
| Temporal resolution (ms) | 232 | | |
| MRI scanner | MAGNETOM Free.Max, Siemens Healthineers, Forchheim, Germany | | |
| Field strength (Tesla) | 0.55 | | |

Functional 2D balanced steady-state free precession (bSSFP) and morphological periodically rotated overlapping parallel lines with enhanced reconstruction (PROPELLER) MRI-Sequence parameters.

[a]Short inversion time inversion recovery (STIR).

[b]Interpolated from 128 × 128. Temporal resolution is only applicable for functional imaging. Units are provided in parentheses for all parameters.

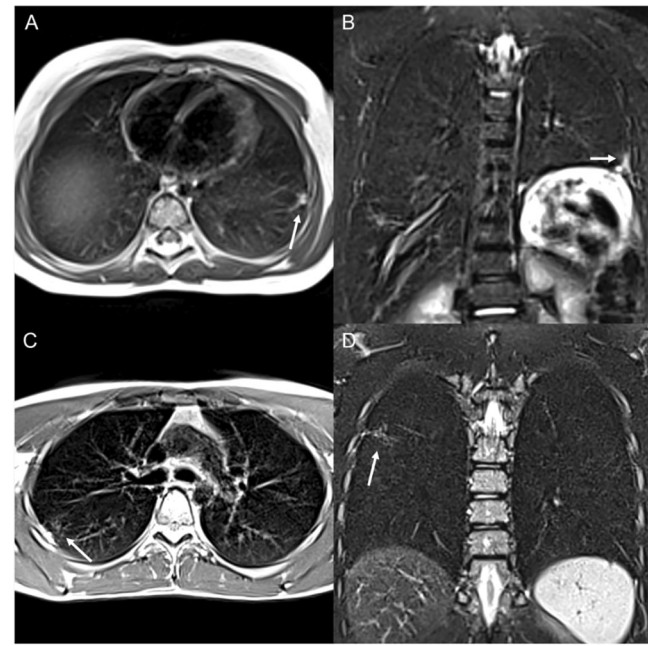

**Fig. 2 | Morphological lung changes detected by LF-MRI.** Two patients exhibited morphological lung changes in proton density-weighted transverse (**A**, **C**) and T2-weighted, fat-saturated coronal sequences (**B**, **D**), using periodically rotated over-lapping parallel lines with enhanced reconstruction (PROPELLER) readout. One patient with acute lymphoblastic leukemia showed linear atelectasis in the left lower lobe (**A**, **B**), while another patient exhibited dystelectasis in the apical right lower lobe (**C**, **D**). Neither reported any clinical symptoms.

functional MRI parameters do not generally differ between solid tumors and systemic diseases such as ALL.

As early as 1986, Miller, R. W. et al. pointed out in their research work that there was still a high incidence of pulmonary dysfunction in those individuals in their study cohort, who either received no radiation or none directed to the thorax. They emphasized that this is an important finding since it would suggest that thoracic radiation is not the only factor contributing to the development of pulmonary function test abnormalities in long-term survivors of childhood cancer[23]. Bleomycin was the first chemotherapy drug shown to cause lung injury[24]. Also, other chemotherapeutic medications, such as carmustine and lomustine, cyclophosphamide, melphalan, or methotrexate and vinblastine may cause pulmonary defects[25–31]. The results of our pilot study are in line with these early observations and, owed to the high sensitivity of our examination, we here visualized functional changes at comparably early time points after treatment. The abnormalities we described regarding the pulmonary ventilation and, at later stages, abnormal lung perfusion could represent an early stage of

potential obstructive lung disease. This hypothesis may also reflect an "accelerated aging" phenotype in these individuals as described by others[32,33]. Like other reports, our study underlines the fact that impaired pulmonary function seems to be multifactorial and present also in survivors not exposed to irradiation, bleomycin or hematopoietic stem cell transplantation[34]. The toxic effect of bleomycin is primarily caused by damage to the small blood vessels and is well known[24]. Although mechanisms e.g., bleomycin-induced toxicity are not necessarily transferable to other types of chemotherapy, it is conceivable that small vessel damage may be an important feature of cytotoxic lung injury in general. Here, PREFUL MRI on a low-field system has proven to be a highly sensitive, safe, non-invasive and easy to use method for detecting these perfusion defects. In addition, we were able to identify temporally preceding ventilation defects. Clinical symptoms for pulmonary or cardiac late effects may occur after a long period of asymptomatic disease. Consequently, a more precise and

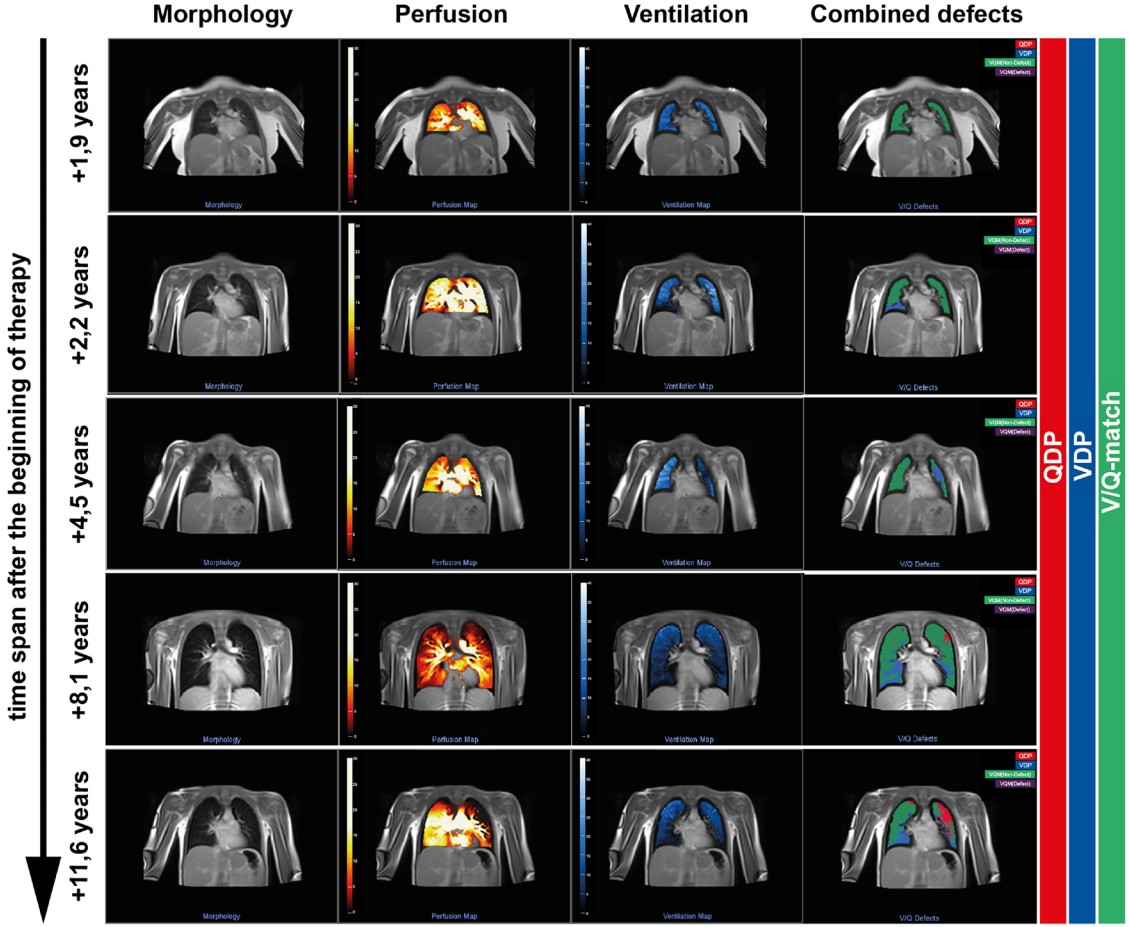

**Fig. 3 | Presentation of ventilation and perfusion matched image data (PREFUL-MRI) of five patients at different time points in the follow-up care after treatment for ALL.** From left to right alongside native images, representative color-coded images from functional lung MRI (perfusion and ventilation map, combined defects) show ventilation defects (blue), perfusion defects (red), and ventilation-perfusion (V/Q) match (V/QM-non-defect, green) in the combined defects map (right). In the patients with a longer time interval since the start of the therapy, both ventilation (blue) and perfusion (red) defects are demarcated, whereas in the patient 1, 9 years after the start of therapy (upper row), the lung tissue still appears defect-free.

early detection of these asymptomatic phases of dysfunction is necessary. As only one patient reported clinical symptoms, this could underline the hypothesis that low-field MRI with PREFUL imaging is a method for timely detection of asymptomatic ventilation deficits before structural damage leads to clinical complaints and perfusion failure.

Our study had some limitations that must be considered when evaluating the results. Firstly, it was a pilot project with a comparably small number of participants. Nevertheless, our analyses yielded highly significant results especially concerning the correlation between the time interval after initiation of treatment for ALL and the degree of the emerging V/Q-mismatch. Comparative imaging examinations in healthy children and adolescents without a clear indication were not performed for ethical reasons. We therefore decided to examine three study participants who had ALL and for whom allogeneic SCT was indicated at a time point before and after stem cell transplantation. Thus, we ensured internal validation of our measurement method and we were unable to detect any significant differences in the close temporal relationship between the two measurements (Table 3). This speaks against a random falsification of the MRI measurements by uncontrollable confounding variables. No significant correlation between the age at time of diagnosis or age at time of examination and the total lung perfusion, ventilation defects or non-defect lung tissue was found. Notably, younger patients at the date of MRI examination had no decreased measurements of non-defect lung tissue due to breathing difficulties or abnormalities which underlines the quality of PREFUL MRI. This method can easily be

repeated on the same patient and can be integrated into daily clinical practice without any special preparations.

## Conclusion
The results of our study show that modern imaging tools are needed, not only for documentation, but also to establish examination methods that can evaluate the effects of health prevention programs and intervention studies based e.g., on the use of early physical exercise during primary therapy and aftercare. The prevention of pulmonary injury is a high priority due to the limited treatment options for chronic lung diseases[4]. Therefore, any additional damage to the lung should be avoided throughout a survivor's life. Our results highlight the importance of preventive measures (e.g., avoiding smoking or workplace-related noxious substances) in this patient group. Pneumococcal and influenza vaccinations should be considered in survivors with established pulmonary disease. In general, prevention and treatment options are limited but the field is evolving quickly. Moreover, through the visualization of pulmonary toxicities, our results could also fuel discussions about a faster replacement of conventional chemotherapeutic regimen by supposedly less toxic immunotherapies or cell therapies such as the primary use of antibodies or CAR-T cells. It appears conceivable that a proportion of patients could benefit from targeting of pathways involved in lung fibrosis, in particular through inhibition of transforming growth factor beta stimulated collagen production, the production of fibrogenic mediators such as transforming growth factor beta, and inflammatory mediators such as tumor necrosis factor alpha and IL-1β[35–37].

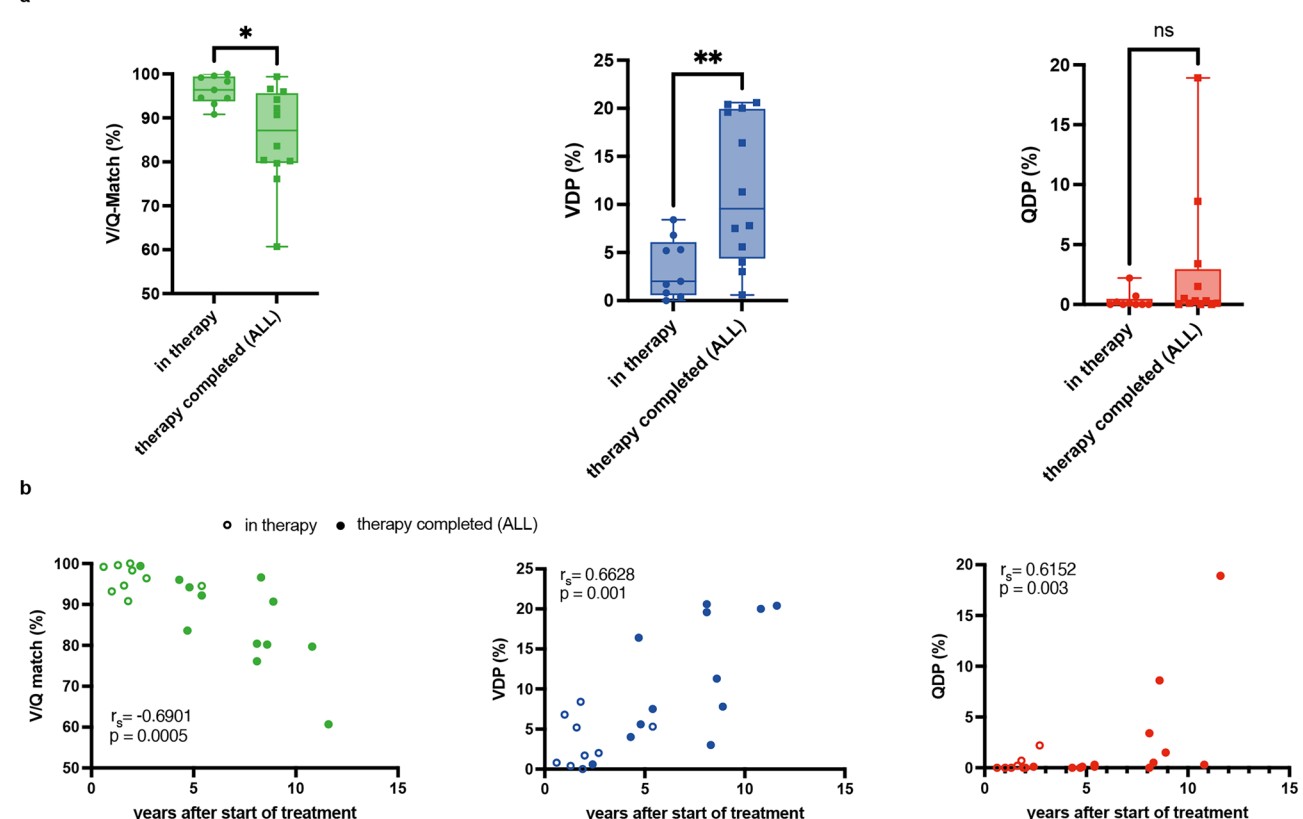

**Fig. 4 | Time dependance of functional changes a** Boxplot-diagrams show the comparison of low-field-strength MRI parameters with respect to the progress of the oncological treatment for ALL (*n* = 21). Patients after completion of the oncological treatment ("therapy completed", *n* = 12) show a significant lower Ventilation-Perfusion-Match (V/Q-Match in %, colored green; independent sample two-tailed *t*-test *p* = 0.0137) and higher whole-lung Ventilation Defect Percentage (VDP in %, colored blue; independent sample two-tailed *t*-test *p* = 0.0079) compared to patients "in therapy" (*n* = 9). No significant differences regarding the whole-lung Perfusion Defect Percentage (QDP in %, colored red; two-tailed Mann–Whitney *U*-Test *p* = 0.1407) between these two groups is observed. **b** A negative and statistically significant correlation between the time span after start of the treatment (in years) and V/Q match (in %) is shown on the left side. Furthermore, significant correlation analysis for both VDP (in %) and QDP (%) and the time span (years) are demonstrated. rs = Spearman-coefficient, * = p < 0.05, ** = p < 0.01), ns = not significant. The error bars in the diagrams represent the respective minimum and maximum values. No adjustments for multiple comparisons were made.

## Table 3 | Repeated MRI examinations for intern validation

| Characteristic | Participant-ID MinimALL_1 | Participant-ID MinimALL_14 | Participant-ID MinimALL_17 |
|---|---|---|---|
| Number of MRI-measurements | 2 | 2 | 2 |
| Time span after initial beginning of therapy until MRI examination (d) | | | |
| before allogeneic SCT | 203 | 990 | 1972 |
| after allogeneic SCT | 589 | 1111 | 2084 |
| Interval between MRI examinations (d) | 386 | 121 | 112 |
| Mean VQ-Non defect (%) | 99.3 ± 0.1 | 96.2 ± 0.2 | 96.1 ± 1.6 |
| Mean VDP (%) | 0.7 ± 0.1 | 3.0 ± 1.0 | 3.8 ± 1.5 |
| Mean QDP (%) | 0 | 1.1 ± 1.1 | 0.5 ± 0.5 |

The MRI data of 3 individuals before and after allogeneic stem cell transplantation are presented. The mean values for the variables VQ-Non defect, VDP and QDP hardly differ between the individual examinations for the same patients despite the time difference. Data for VQ-Non defect, VDP and QDP are means ± SDs.

In summary, free-breathing low-field MRI with PREFUL imaging is safe and has proven to be an effective and easy to use tool in detecting potential subclinical organ damage of the lung after oncological treatment in pediatric patients. To the best of our knowledge, such functional deficits in MRI imaging have not yet been described elsewhere. Whether these are reversible incidental findings or the onset of a relevant pulmonary disease must be the subject of further research projects. Essential subsequent larger studies will be initiated to corroborate these present findings based on higher patient numbers and the inclusion of adult pediatric cancer survivors as well as age matched controls.

## Data availability

The raw, individual and identifiable patient MRI data are protected and are not available due to data privacy laws. The source data for Fig. 4 is provided in the Supplementary Data file 1. Supplementary Data file 2 provides the

clinical characteristics and results of the PREFUL-MRI examination of the study participants.

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

## Acknowledgements
We thank the Imaging Science Institute Erlangen for providing us with measurement time and technical support.

## Author contributions
A.D., M.H., E-M.W., A.K. and F. K. designed and performed the clinical study. F.A., N.M., F.H., I.S. and N.N-B. supported the recruitment of study

participants. A.P.R. and J.W. provided essential support to the clinical study. N.B., M.U., R.H., A.M.N., S.S., R.G., J.V-C and A.V. provided scientific background and technical support. A.D., E-M.W., N.B. and M.H. analyzed and interpreted the data. A.K. and F.K. conceived and supervised the project. A.D., E-M.W., M.H. and A.K. wrote the first draft of the manuscript. All authors edited and approved the final draft.

## Funding

## Competing interests
M.H., M.U., and R.H. are part of the speakers' bureau of Siemens Healthineers GmbH. R.G. is an employee of Siemens Healthineers (Erlangen, Germany). J.VC. and A.V. are shareholders in BioVisioneers GmbH, a company with an interest in pulmonary MRI methods. F.K. is a member of the advisory board of iThera Medical GmbH, Munich, Germany, received travel support from iThera Medical GmbH, Germany, UCB Pharma, Germany and Sanofi Aventis, Germany. F.K. reports lecture fees from Sanofi Aventis, Germany and Siemens Healthcare GmbH, Germany. A.P.R. received travel support from iThera Medical GmbH, Germany and Sanofi Aventis, Germany. N.B. is supported by the IZKF Clinician Scientist Program and the IZKF Laboratory Rotation Program of the Medical Faculty of the Friedrich-Alexander-Universität Erlangen-Nürnberg.All other authors declare no competing interest.

## Additional information

**Alexander Dierl** [1,8] ✉**, Maximilian Hinsen** [2,8]**, Eva-Maria Wild** [1]**, Nadine Bayerl** [2]**, Rafael Heiss** [2]**, Armin M. Nagel** [2,3]**, Sandy Schmidt**[2]**, Robert Grimm**[4]**, Jens Vogel-Claussen**[5,6]**, Andreas Voskrebenzev**[5,6]**, Nora Naumann-Bartsch** [1]**, Felix Anderheiden**[7]**, Felix Huber**[7]**, Nicolas Mueller**[7]**, Isabelle Schoeffl**[7]**, Joachim Woelfle**[1]**, Michael Uder**[2]**, Adrian P. Regensburger** [1]**, Ferdinand Knieling** [1] **& Axel Karow**[1]

[1]Department of Pediatrics and Adolescent Medicine, University Hospital Erlangen, Friedrich-Alexander-Universität (FAU) Erlangen-Nürnberg, Erlangen, Germany. [2]Institute of Radiology, University Hospital Erlangen, Friedrich-Alexander-Universität (FAU) Erlangen-Nürnberg, Erlangen, Germany. [3]Division of Medical Physics in Radiology, German Cancer Research Center, Heidelberg, Germany. [4]Research & Clinical Translation, Magnetic Resonance, Siemens Healthineers AG, Erlangen, Germany. [5]Institute of Diagnostic and Interventional Radiology, Hannover Medical School, Hannover, Germany. [6]Biomedical Research in Endstage and Obstructive Lung Disease Hannover (BREATH), German Center for Lung Research (DZL), Hannover, Germany. [7]Department of Pediatric Cardiology, University Hospital Erlangen, Friedrich-Alexander-Universität (FAU) Erlangen-Nürnberg, Erlangen, Germany. [8]These authors contributed equally: Alexander Dierl, Maximilian Hinsen. ✉e-mail: alexander.dierl@uk-erlangen.de

