## [Transparent Peer Review file · Communications Medicine]

Free-breathing phase-resolved functional lung (PREFUL) low-field magnetic resonance imaging (LF-MRI) of pulmonary dysfunction after surviving childhood cancer

Corresponding Author: Dr Alexander Dierl

Version 0:

Reviewer comments:

Reviewer #1

(Remarks to the Author)
Peer Review Report

This manuscript addresses an important clinical issue: the detection of pulmonary late effects in survivors of childhood acute lymphoblastic leukemia and Hodgkin's disease. The use of free-breathing phase-resolved functional lung (PREFUL) low-field MRI represents a novel and non-invasive approach with clear clinical relevance. The feasibility of this imaging method is convincingly demonstrated.

Strengths

The study targets a significant gap in survivorship research by focusing on early pulmonary dysfunction in a vulnerable population.

PREFUL MRI offers a radiation-free, contrast-free technique, making it particularly appropriate for pediatric follow-up.

Results highlight the potential for subclinical detection of ventilation and perfusion defects before clinical symptoms develop.

Limitations

The pilot nature of the study, small sample size, and cross-sectional design limit generalizability.

Absence of a healthy control group makes it challenging to attribute functional defects exclusively to prior cancer therapy.

Longitudinal follow-up would be required to confirm whether observed changes progress over time and to strengthen causal inferences.

Clarity and Presentation

Although the scientific content is relevant, the manuscript requires revision, as the current writing style is somewhat cumbersome and occasionally redundant, which hinders readability.

The background section could be shortened to enhance focus.

I suggest to split your long narrative into the usual sections: Background → Objective → Methods → Results → Discussion → Conclusion.

Figures are valuable but would benefit from more detailed legends, particularly regarding MRI acquisition parameters and analysis.

Others

Standardized terminology (e.g., always "VDP, QDP, V/Q match" instead of repeating the full term each time).

Some statistical phrasing (e.g., "highly inversely significant") should be revised to standard terminology.

Add missing commas, broke up very long sentences.

Reviewer #2

(Remarks to the Author)

This manuscript investigates lung function and morphology in childhood and adolescent cancer survivors using phase-resolved functional lung low-field (PREFUL) free-breathing MRI. This pilot study includes 27 pediatric patients who were

treated for either acute lymphoblastic leukemia or Hodgkin's disease. Despite the absence of clinical respiratory symptoms, imaging studies revealed significant ventilation and perfusion defects that appeared to have progressively worsened since treatment. The authors propose that PREFUL LF MRI could fill an important diagnostic gap by detecting early, non-radiative subclinical lung dysfunction in this vulnerable population.

These conclusions are novel and very interesting, and of great interest to pediatric oncology and radiology, as well as pediatric pulmonology. Although late pulmonary sequelae are known in childhood cancer survivors, sensitive early diagnostic techniques, particularly radiation-free functional imaging without contrast, are still underdeveloped. The use of PREFUL LF MRI in this population could therefore represent an important advance in early detection. Comparable studies using functional lung MRI exist. However, these are limited in the pediatric setting, which underscores the originality of this study.

The main conclusions of this study are that childhood cancer survivors, even without clinical respiratory symptoms, exhibit significant time-dependent pulmonary ventilation and perfusion defects. The study suggests that PREFUL LF MRI is a promising, non-invasive, and radiation-free imaging modality that can sensitively identify early subclinical lung damage in this high-risk population, potentially enabling earlier interventions and improved long-term monitoring.

These conclusions are novel and very interesting, and of great interest to pediatric oncology and radiology, as well as pediatric pulmonology. Although late pulmonary sequelae are known in childhood cancer survivors, sensitive early diagnostic techniques, particularly radiation-free functional imaging without contrast, are still underdeveloped. The application of PREFUL LF MRI in this population could therefore represent an important advance in early detection strategies. Comparable studies of functional lung MRI exist, but are limited in the pediatric setting, which underscores the originality of this study.

Although the study is compelling as a pilot demonstration of feasibility and preliminary results, it is limited by the small sample size, the heterogeneity of the cohort, and the lack of healthy controls or longitudinal data to confirm progression. Validation of the imaging biomarkers against clinical data or established pulmonary function tests would also strengthen the link between MRI findings and clinical relevance.

The results of this and other, slightly larger studies could influence thinking regarding childhood cancer survival by incorporating advanced functional lung imaging into routine follow-up protocols. Furthermore, depending on availability and cost, the study could inspire similar applications of low-field MRI in other chronic childhood lung diseases and contribute to wider acceptance of radiation-free imaging biomarkers in clinical practice.

Version 1:

Reviewer comments:

Reviewer #1

(Remarks to the Author)

I thank the authors for the revisions made. I have no further requests.

Reviewers' comments:

Reviewer #1 (Remarks to the Author):

Peer Review Report

This manuscript addresses an important clinical issue: the detection of pulmonary late effects in survivors of childhood acute lymphoblastic leukemia and Hodgkin's disease. The use of free-breathing phase-resolved functional lung (PREFUL) low-field MRI represents a novel and non-invasive approach with clear clinical relevance. The feasibility of this imaging method is convincingly demonstrated.

Strengths

The study targets a significant gap in survivorship research by focusing on early pulmonary dysfunction in a vulnerable population.

PREFUL MRI offers a radiation-free, contrast-free technique, making it particularly appropriate for pediatric follow-up.

Results highlight the potential for subclinical detection of ventilation and perfusion defects before clinical symptoms develop.

AUTHORS' RESPONSE:

Thank you, we absolutely agree with the reviewer and we very much appreciate this encouraging comment.

Limitations

The pilot nature of the study, small sample size, and cross-sectional design limit generalizability.

Absence of a healthy control group makes it challenging to attribute functional defects exclusively to prior cancer therapy.

Longitudinal follow-up would be required to confirm whether observed changes progress over time and to strengthen causal inferences.

AUTHORS' RESPONSE:

We are grateful to the reviewer for addressing this important issue. As also stated in the discussion / conclusion section, the sample size of this pilot study is indeed limited. However, especially against this background, the key observation of time-dependent pulmonary ventilation and perfusion defects appears significant and convincing.

We also agree that a comparison with healthy controls would be desirable. However, including an age-matched healthy control group was not feasible in the frame of this pilot study due to ethical restrictions, which did not allow the examination in healthy children.

In addition, longitudinal data would certainly be valuable. Realization of a study with repeated measurements including the same individuals over time appeared hardly feasible at this level since pulmonary alterations obviously develop over many years after therapy.

Consequently, based on the results of the MinimALL pilot study, we are currently initiating a subsequent large research project with 150 participants also including adult pediatric-cancer survivors as well as an age-matched control group.

Through this upcoming research project, we aim to raise the level of evidence with a larger sample size and examine patients with a longer time-span after initial therapy in comparison to normal controls. Ultimately, integration of these examinations into regular follow up care would enable individual longitudinal data.

Clarity and Presentation

Although the scientific content is relevant, the manuscript requires revision, as the current writing style is somewhat cumbersome and occasionally redundant, which hinders readability.

The background section could be shortened to enhance focus.

I suggest to split your long narrative into the usual sections: Background → Objective → Methods → Results → Discussion → Conclusion.

Figures are valuable but would benefit from more detailed legends, particularly regarding MRI acquisition parameters and analysis.

AUTHORS' RESPONSE:

We'd like to thank the reviewer for this remark and we have now revised the manuscript accordingly to improve clarity and readability. The writing style has been streamlined to avoid redundancy and the structure has been reorganized into the conventional sections (Background, Objective, Methods, Results, Discussion, Conclusion) to enhance focus and reading flow.

Figure legends have been expanded and now include detailed information on MRI acquisition parameters and analysis procedures as suggested.

Others

Standardized terminology (e.g., always "VDP, QDP, V/Q match" instead of repeating the full term each time). Some statistical phrasing (e.g., "highly inversely significant") should be revised to standard terminology. Add missing commas, broke up very long sentences.

AUTHORS' RESPONSE:

Thank you. Terminology has now been standardized throughout the manuscript. Statistical phrasing has been revised to follow standard conventions. Furthermore, we checked for spelling errors and restructured the manuscript regarding the readability.

Reviewer #2 (Remarks to the Author):

This manuscript investigates lung function and morphology in childhood and adolescent cancer survivors using phase-resolved functional lung low-field (PREFUL) free-breathing MRI. This pilot study includes 27 pediatric patients who were treated for either acute lymphoblastic leukemia or Hodgkin's disease. Despite the absence of clinical respiratory symptoms, imaging studies revealed significant ventilation and perfusion defects that appeared to have progressively worsened since treatment. The authors propose that PREFUL LF MRI could fill an important diagnostic gap by detecting early, non-radiative subclinical lung dysfunction in this vulnerable population.

These conclusions are novel and very interesting, and of great interest to pediatric oncology and radiology, as well as pediatric pulmonology. Although late pulmonary sequelae are known in childhood cancer survivors, sensitive early diagnostic techniques, particularly

radiation-free functional imaging without contrast, are still underdeveloped. The use of PREFUL LF MRI in this population could therefore represent an important advance in early detection. Comparable studies using functional lung MRI exist. However, these are limited in the pediatric setting, which underscores the originality of this study.

The main conclusions of this study are that childhood cancer survivors, even without clinical respiratory symptoms, exhibit significant time-dependent pulmonary ventilation and perfusion defects. The study suggests that PREFUL LF MRI is a promising, non-invasive, and radiation-free imaging modality that can sensitively identify early subclinical lung damage in this high-risk population, potentially enabling earlier interventions and improved long-term monitoring.

These conclusions are novel and very interesting, and of great interest to pediatric oncology and radiology, as well as pediatric pulmonology. Although late pulmonary sequelae are known in childhood cancer survivors, sensitive early diagnostic techniques, particularly radiation-free functional imaging without contrast, are still underdeveloped. The application of PREFUL LF MRI in this population could therefore represent an important advance in early detection strategies. Comparable studies of functional lung MRI exist, but are limited in the pediatric setting, which underscores the originality of this study.

AUTHORS' RESPONSE:

Thank you very much indeed, we really appreciate these inspiring statements.

Although the study is compelling as a pilot demonstration of feasibility and preliminary results, it is limited by the small sample size, the heterogeneity of the cohort, and the lack of healthy controls or longitudinal data to confirm progression. Validation of the imaging biomarkers against clinical data or established pulmonary function tests would also strengthen the link between MRI findings and clinical relevance.

AUTHORS' RESPONSE:

We are grateful to the reviewer for this relevant remark. As also stated in the discussion / conclusion section, we fully acknowledge the limitations related to the small sample size, cohort heterogeneity and absence of healthy controls or longitudinal data. The sample size of this pilot study is indeed limited. However, especially against this background, the key observation of time-dependent pulmonary ventilation and perfusion defects appears significant and, as also addressed in the reviewer's comment, compelling. We also agree that a comparison with healthy controls would be desirable. However, including an age-matched healthy control group was not feasible in the frame of this pilot study due to ethical restrictions, which did not allow the examination in healthy children. In addition, longitudinal data would certainly be valuable. Realization of a study with repeated measurements including the same individuals over time appeared hardly feasible at this level since pulmonary alterations obviously develop over many years after therapy.

Consequently, based on the results of the MinimALL pilot study, we are currently initiating a subsequent large research project with 150 participants mainly after ALL therapy also including adult pediatric-cancer survivors as well as an age-matched control group.

Through this upcoming research project, we aim to raise the level of evidence with a larger sample size and examine patients with a longer time-span after initial therapy in comparison to normal controls. Furthermore, validation of imaging biomarkers against established pulmonary function tests, including spirometry and cardiopulmonary exercise testing (spiroergometry), is planned to strengthen the correlation between MRI-derived parameters and clinical relevance.

The results of this and other, slightly larger studies could influence thinking regarding childhood cancer survival by incorporating advanced functional lung imaging into routine follow-up protocols. Furthermore, depending on availability and cost, the study could inspire similar applications of low-field MRI in other chronic childhood lung diseases and contribute to wider acceptance of radiation-free imaging biomarkers in clinical practice.